# Effects of Vapor Pressure of Desiccant Solution on Mass Transfer Performance for a Spray-Bed Absorber

**Hung-Ta Wu** [1],* and **Chin-Chun Chung** [2]

1 Department of Chemical and Materials Engineering, National Ilan University, No. 1, Sec. 1, Shennong Rd., Yilan 260007, Taiwan
2 Department of Chemical Engineering, Army Academy, No. 750, Longdong Rd., Chung-Li District, Taoyuan 320316, Taiwan; doctorjons@aaroc.edu.tw
* Correspondence: htwu@niu.edu.tw

**Abstract:** The depression in vapor pressure caused by adding desiccant to liquid water can be regarded as the driving force for the dehumidification process. The vapor pressure depends on the temperature and the concentration. Therefore, the purpose in this study is to discuss the mass transfer performance affected by operating variables and to show that the vapor pressure is a key factor affecting the mass transfer performance for absorbing water vapor by triethylene glycol (TEG) solution. The experimental results showed that the mass transfer coefficients were decreased with increases in the temperature and increased with increases in the concentration, respectively, while the mass transfer coefficients were increased with increases in the vapor pressure depression. Although both the average error is within 5% among the mass transfer correlation involving the vapor pressure and that involving the temperature and the concentration in predicting the mass transfer coefficient, there are just two terms, those are vapor pressure and fluid flow rate, associated with operating variables used in the mass transfer correlation. The depression in vapor pressure was not only proved to be the driving force for absorbing water vapor by a desiccant solution, but also a key factor affecting the mass transfer performance.

**Keywords:** mass transfer; vapor pressure; driving force; absorption

## 1. Introduction

The vapor pressure will be depressed as addition of a desiccant into pure water. The vapor pressure is a key factor for choosing a desiccant solution to use in the dehumidification system. Generally speaking, the vapor pressure of a desiccant solution is extremely low, meaning that only a small amount water vapor on the gas–liquid interface, in the process of absorbing water vapor by a desiccant solution. The vapor pressure difference between the interface and the gas phase will be formed to make water vapor flowing from bulk gas phase to the interface and absorbed by a desiccant solution. Therefore, the depression in vapor pressure or vapor pressure difference can be regarded as the driving force in a dehumidification process. Mentioned above, the vapor pressure of a desiccant solution is an important variable dominating the mass transfer performance for moisture absorbed by a desiccant solution in an absorber. However, discussions on the effect of the vapor pressure on the performance of this kind absorber were rare in the literature. The vapor pressures were measured under the controlled temperature and concentration to investigate the effect of vapor pressure on the performance and establish the mass transfer correlation for absorption of water vapor by the TEG solution in an absorber.

As shown in Table 1, the common absorption devices used in separation process includes packed-bed, spray-bed, falling-film absorber, and membrane contactor. Since the interfacial area between gas and liquid phases can be provided by the packing naturally, the studies on packed-bed absorber account for more than half. The absorption tower is one of the main devices for the heat pump system, and the efficiency of the heat pump depends

on the mass transfer performance of the absorption tower. Therefore, salt solutions, such as calcium chloride, lithium chloride, and lithium bromide, were used to absorb water vapor in the packed-bed absorber to discuss the mass transfer performance [1–4]. In addition to water vapor absorbed by a desiccant solution, the studies associated with carbon dioxide removed by the monoethanolamine solution and the mixture of monoethanolamine and glycerol solutions were conducted by Park and Øi [5] and Valeh-e-Sheyda and Barati [6], respectively. The advantage for using structured packing is that the packing is removed from the packed bed easily, but the disadvantage is that cleaning the packing is difficult. In order to discuss the performance of the tower packed with structured packing, water vapor absorbed by the TEG solution was conducted by Abdul-Wahab et al. [7] and Elsarrag [8] and that absorbed by the LiCl solution was conducted by Tang et al. [9] To improve the problem of solution dilution, caused by system freeze, a cross flow regenerator with heat supplying to air in the packed bed was established by Huang et al. [10] The rotating packed bed (RPB) employing a centrifugal force to enhance the heat and mass transfer processes was applied by Gu and Zhang [11] and Liu et al. [12] to remove water vapor and sulfur dioxide, respectively. The gravity factor or the rotational speed is a key factor for the RPB absorber.

The interfacial area between gas and liquid phases in the spray-bed is determined by the atomized effect. The better the atomized effect, the larger the interfacial area. Both glycol and salt solutions can be used as a working solution in a spray bed. For example, Tanda et al. [13] used polyethylene glycol (PEG) solution as a desiccant solution to develop mass transfer correlation for a spray column. Huang et al. [14] used $CaCl_2$ aqueous solution as a working solution to model heat and mass transfer behaviors for a heating tower. Similar to the spray droplet, the model for carbon dioxide captured by a stationary single droplet was developed by Chen et al. [15] to analyze effects of temperature and time on the absorption rate and the concentration of carbon dioxide. Compared with the packed-bed absorber, there are fewer studies on the spray-bed absorber. Gandhidasan et al. [16] described that the vapor pressure in a dehumidification system could be equilibrated by transferring water vapor from gas phase to liquid phase, thus the vapor pressure difference between gas phase and desiccant solution could be regarded as the driving force in the dehumidification system. However, the vapor pressure is not found in the variable column of Table 1. The driving force, the depression in vapor pressure, for absorption of water vapor from air should be regarded as a variable to discuss the mass transfer performance. Therefore, the purpose in this study was to discuss the mass transfer performance and the correlation associated with the temperature, the concentration and the vapor pressure of the desiccant solution to show that the vapor pressure is a key factor affecting the mass transfer performance of an absorption tower. Combination of the spray bed and film coil for designing internally heated regenerator was modeled by Song et al. [17] to show that the regenerator performance of the internally heated regenerator was affected by the air flow rate, heating water flow rate, and inlet temperature of heating water.

**Table 1.** The searched studies associated with the absorption performance affected by operating variables similar to this study.

| Device Type | Absorbent | Absorbate | Variable | Response | Reference |
|---|---|---|---|---|---|
| packed bed | $CaCl_2$ | $H_2O$ | $T_G$ | COP<br>air mass ratio | [1] |
| packed bed | LiCl | $H_2O$ | G, L, Q, $\varepsilon$ | evaporation rate | [2] |
| packed bed | LiCl<br>LiBr<br>KCOOH | $H_2O$ | $T_G$, $T_L$, $G_s$, $L_s$, H, $C_L$ | removal rate<br>regeneration rate<br>pressure drop | [3] |
| packed bed | LiCl<br>LiBr<br>$CaCl_2$ | $H_2O$ | $T_G$, $T_L$, H, G, L | removal rate<br>and efficiency | [4] |
| packed bed | MEA | $CO_2$ | packing type | pressure drop<br>operating cost | [5] |
| packed bed | MEA-glycerol | $CO_2$ | $T_L$, G, $C_L$ | K<br>$\varepsilon$ | [6] |
| structured packed bed | TEG | $H_2O$ | G, L, $T_G$, $T_L$, H, $C_L$ | removal rate<br>$\varepsilon$ | [7] |
| structured packed bed | TEG | $H_2O$ | G, L, H, $T_L$, $C_L$ | evaporation rate<br>$\varepsilon$ | [8] |
| structured packed bed | LiCl | $H_2O$ | G, L | K | [9] |
| structured packed bed<br>(heating tower) | glycol solution | $H_2O$ | $G_s$, $L_s$, $T_G$, $T_L$, H, $C_L$ | removal rate<br>$\varepsilon$<br>thermal efficiency | [10] |
| rotating packed bed | LiCl | $H_2O$ | g, $G_s$, $L_s$, inlet parameters of air and liquid | removal rate<br>H, K, $\varepsilon$ | [11] |
| rotating packed bed | ([CPL][TBAB])IL | $SO_2$ | r, $T_L$, G, L | $\varepsilon$ | [12] |
| spray bed | polyethylene glycol | $H_2O$ | $G_s$, $L_s$, CL, $T_L$, $d_n$ | K | [13] |
| spray bed | $CaCl_2$ aqueous | $H_2O$ | G, L, $T_G$, H, $T_L$, $C_L$ | E | [14] |
| spray/single droplet | selexol, rectisol, water | $CO_2$ | $T_L$, P, ab | absorption rate<br>concentration | [15] |
| spray bed<br>(regenerator) | glycerol solution | $H_2O$ | $L_H$, $T_w$, G | removal rate<br>latent heat ratio<br>$\varepsilon$ | [17] |
| falling film absorber | TEG | $H_2O$ | G, L, $T_w$, $T_L$ | COP | [18] |
| falling film absorber | limestone slurry | $SO_2$ | pH, $C_L$ | $\varepsilon$ | [19] |
| falling film absorber | LiCl | $H_2O$ | G, H, W | humidity ratio<br>concentration | [20] |
| membrane contactor | LiCl | $H_2O$ | me | $\varepsilon$ | [21] |
| membrane contactor | LiCl | $H_2O$ | H, $C_L$, G/L | $\varepsilon$, removal rate<br>cooling capacity<br>COP | [22] |
| hollow fiber<br>membrane contactor | diethanolamine<br>solution | $CO_2$<br>$H_2S$ | G, $C_f$, $F_w$, $h_M$ | recovery<br>selectivity | [23] |

COP: coefficient of performance; MEA: monoethanolamine; TEG: triethylene glycol; ([CPL] [TBAB]) IL: Caprolactam tetrabutyl ammonium bromide; $T_G$: gas temperature; $T_L$: liquid temperature; G: gas flow rate; L: liquid flow rate; $G_S$: gas flow flux; $L_S$: liquid flow flux; $C_L$: liquid concentration; H: humidity; Q: heat input; g: gravity factor; r: rotational speed; $d_n$: nozzle diameter; P: pressure; ab: absorbent type; $L_H$: heating water flow rate; $T_w$: water temperature; pH: pH value; W: channel width; me: membrane type; $C_f$: feed concentration; $F_w$: wetting fraction; $h_M$: module height; K: mass transfer coefficient; H: heat transfer coefficient; E: enthalpy transfer coefficient; $\varepsilon_H$: heat effectiveness; $\varepsilon$: removal efficiency.

The advantage for using the falling film absorber as a removal device is the lower pressure drop. For example, Kumar et al. [18] used the TEG solution as a working solution in the falling film absorber to absorb water vapor from air to discuss the performance of a hybrid liquid desiccant system. Recently, Zhou and Zhang [19] used the limestone slurry to remove sulfur dioxide in a falling film absorber to discuss the removal efficiency affected by the pH value and the $SO_2$ concentration. Similar to the principle of the falling

film absorber, the internally cooled/heated dehumidifier/regenerator was established by Yin et al. [20] to improve the dehumidification/regeneration performance.

The principle of the indirect air–liquid membrane contactor is that the porous membrane acts as a barrier between liquid desiccant and air and the water vapor can pass through the membrane but liquid cannot pass. To overcome the carryover carried out from the absorber, the membrane contactors using microporous semipermeable hydrophobic membranes was applied by Das and Jain [21,22] to discuss the effects of operating variable on the removal effectiveness for absorption of water vapor by lithium chloride solution. The microporous hollow fiber membrane contactor for gas absorption and stripping is still a relatively new concept in comparing with the usual membrane contactor. In order to analyze the simultaneous absorption of carbon dioxide and hydrogen sulfide by an aqueous solution of diethanolamine, a mathematic model assuming that all reactions were reversible in the liquid phase as well as the wetted parts of the membrane pores for a microporous hollow fiber membrane contactor was developed by Keshavarz et al. [23].

The establishment of the mass transfer correlation can be divided into two categories in the literature. One is based on the operating variables affecting the mass transfer coefficient and the characteristics of fluid to obtain a dimensionless correlation through the dimensional analysis method. The important variables are regressed nonlinearly without considering the dimension to obtain mass transfer correlation is the other one. As shown in Table 2, the dimensionless group, such as Reynolds and Schmidt numbers, were considered to develop the mass transfer correlation by the dimensional analysis. Since the Sherwood number was the ratio of the multiplication of mass transfer coefficient and characteristic length to the diffusion coefficient, the correlation among Sherwood number, Reynolds number, and Schmidt numbers was developed by Yin et al. [20]. The dimensionless group involving the mass transfer coefficient could be modified due to the characteristic of the dimensionless method. For example, the dimensionless group—including the mass transfer coefficient, the molecular weight, the column diameter, the diffusion coefficient, and the gas density—was used by Tanda et al. [13] to develop the mass transfer correlation. In addition to the operating variables affecting mass transfer coefficient, the surface tension was also considered by Tanda et al. [13] due to the spray effect affected by surface tension significantly. The mass transfer coefficient is placed directly in the left of the equal sign, and the dimensionless groups, Reynolds and Schmidt numbers, and the dimension analysis were considered by Hanley and Chen [24] to develop the mass transfer correlations for different packing. Similarly, the mass transfer coefficient was used by Sánchez et al. [25] directly to associate with the dimensionless groups, and the liquid holdup and the packing properties were also used to develop the correlation. The main difference from the packed-bed absorber for the correlation developed by Gu and Zhang [11] was that Grashof number used for a rotating packed-bed absorber.

**Table 2.** Recent studies discussed on mass transfer correlation in the absorber/stripper.

| Type (Absorber Type) | Correlation | Dimensionless Group for Fluid Property | Reference |
|---|---|---|---|
| dimensionless correlation (falling film) | $Sh = 4.513 \times 10^{-3} \cdot \alpha \cdot Re^{1.56} Sc^{0.33}$ | Re, Sc | [20] |
| dimensionless correlation (spray bed) | $\dfrac{K_{mA} M d_c^2}{D_G \rho_G} = 1.4 \times 10^3 Re_G^{8.90} Sc_G^{0.33}$ $Re_L^{-0.40} \left(\dfrac{L}{G}\right)^{8.41} \left(\dfrac{d_n}{d_c}\right)^{2.31} \left(\dfrac{\rho_L}{\rho_G}\right)^{-15.42} \left(\dfrac{L_s^2 d_n}{\sigma_L \rho_L}\right)^{-3.3}$ | Re, Sc | [13] |
| dimensionless correlation (spray bed) | $\dfrac{K_{mA} M d_c^2}{\rho D} = 0.0147 Re^{1.45} Sc^{0.35}$ $\left(\dfrac{L}{G}\right)^{0.23} \left(\dfrac{P_{H_2O}}{P_{TEG}}\right)^{0.41}$ | Re, Sc | this study |

**Table 2.** *Cont.*

| Type (Absorber Type) | Correlation | Dimensionless Group for Fluid Property | Reference |
|---|---|---|---|
| dimensionless correlation (packed bed) | $k_m = \alpha \cdot Re_G Sc_G^{1/3}(\frac{c_L D_L}{d_e})$, $\alpha = 0.00104$ for metal pall rings, $\alpha = 0.00473$ for metal IMTP, $\alpha = 0.0084$ for sheet metal structured packing | Re, Sc | [24] |
| dimensionless correlation (packed bed) | $k_k = 0.1304 \cdot C_V \cdot (\frac{D_G \cdot P}{R \cdot T})$ $(\frac{a}{[\varepsilon(\varepsilon - h_L)]^{0.5}}) \cdot (\frac{Re_G}{K_W})^{-3/4} Sc_G^{2/3}$ | Re, Sc | [25] |
| dimensionless correlation (rotating packed bed) | $\frac{H_{kA}}{\rho_a D_a a_t^2} = 3.88 \times 10^{-4} Re_a^{0.58} Re_s^{0.56} Gr^{0.22}$ | Re, Gr | [11] |
| dimensionless correlation (spray bed) | $\frac{H_{mA}V}{m_s} = 0.5537(\frac{m_s}{m_a})^{-0.6730}$ (winter) | none | [14] |
| | $\frac{H_{mA}V}{m_w} = 0.5828(\frac{m_w}{m_a})^{-0.6957}$ (summer) | none | |
| without considering dimension (structured packed bed) | $h_m = 4.4328 \cdot S_s^{0.73793} \cdot S_a^{0.61805}$ | none | [10] |
| without considering dimension (spray bed) | $h_k = 0.01844 \cdot T_s^{-0.1755} \cdot S_s^{1.0167}$ | none | [17] |

Although the dimensionless form was still held, the dimensionless groups for physical properties of fluid flow were not considered by Huang et al. [14] to regress the relationship between mass transfer coefficient and flow rates. In addition, the correlation for mass transfer coefficient was also developed directly by the operating variables affecting the mass transfer coefficient without considering the dimension. For example, the correlations for the structured packed bed and the spray-bed absorbers were developed by Huang et al. [10] and Song et al. [17], respectively. Although the dimensionless groups, the operating variables, and the packing properties were used to develop the mass transfer correlation mostly, the driving force, the depression in vapor pressure, for the dehumidification process was not seen among these correlations. Therefore, one of purposes in this study was to establish the mass transfer correlation to show that the vapor pressure is a key factor affecting the mass transfer performance in the dehumidification process. Actually, the depression in vapor pressure means that the reduction in vapor pressure of pure water caused by adding desiccant to liquid water. Since the vapor pressure of a pure desiccant approaches zero, the vapor pressure of an aqueous desiccant solution is contributed by water. The vapor pressure of the aqueous desiccant solution can be regarded as the depression in vapor pressure for a certain composition. In addition to the relevant dimensionless groups, the ratio of liquid to gas flow rates and the depression in vapor pressure were used to establish the mass transfer correlation in this study.

## 2. Experimental Section

### 2.1. Experimental Procedure

The vapor pressures of desiccant solution were measured to analyze the mass transfer performance for using the desiccant solution in the dehumidification system qualitatively. Compared with the packed-bed absorber, there are fewer studies on the spray-bed absorber and the advantages including the simple structure, the smaller resistance, and the fewer blocking are for the spray-bed absorber. Therefore, the spray-bed absorber was set in this study. To discuss the mass transfer performance for the spray-bed absorber, the variables—including the gas and liquid flow rates, and the temperature and concentration of the TEG solution—were controlled during experimental runs. In addition, the mass transfer

correlation based on some dimensionless groups and the relevant factors was developed in this study. The literature data were also used to test the correlation.

### 2.2. Measuring System for Vapor Pressure

Since the conventional compressor can be replaced by the absorber and the stripper in the absorption air-conditioning system, the performance of an absorption air-conditioning system will be dominated by the absorption performance for water vapor absorbed by a desiccant solution. In general, the depression in vapor pressure, caused by adding desiccant into liquid water, was regarded as the driving force in the dehumidification process. The measuring system for vapor pressure modified from our previously study was set, as shown in Figure 1, to measure the vapor pressure and the vapor pressure was used for the mass transfer correlation. The steps for measuring the vapor pressure of desiccant solution are listed as follows:

(1) The desiccant solution is placed in STIR.
(2) Liquid nitrogen in the Dewar bottle was used to freeze or solidify the desiccant solution for about 10 min.
(3) The valvesV2 and V4 are closed. The vacuum pump was activated firstly and then the valves V1, V3, V5, and V6 are opened for degassing. (solid state still for the desiccant solution)
(4) After activating the vacuum pump, the system through V1, V3, V5, and V6 is degassed by the vacuum pump.
(5) While reaching the vacuum state, V1 could be closed to maintain the vacuum state of the system through V1, V3, V5, and V6 and vacuum pump could be close to save energy.
(6) The temperature of heating stirrer is increased to defrost the desiccant solution.
(7) Steps 2–6 are repeated three times to remove impurities in the system.
(8) The temperatures of the desiccant solution, 20, 25, 30, and 35 °C, controlled by heating stirrer were used to meet experimental demand. The vapor pressure of the desiccant solution was obtained by a pressure gauge after reaching the equilibrium state between gas and liquid phases.

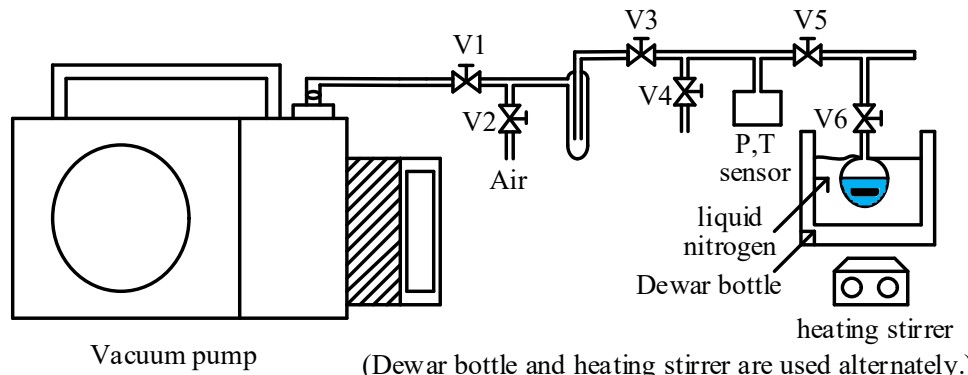

**Figure 1.** Measuring system for vapor pressure.

### 2.3. Spray-Bed Absorber and Operating Procedure

Figure 2 shows a spray-bed absorption system. The air inlet and outlet are located on both sides of the upper parts of absorber and stripper. The cross-sectional area of the air/liquid duct providing fluid flow in the tower is $13 \times 13$ cm$^2$. The solution inlet is located 6 cm below the air outlet, and the solution is sprayed as atomized droplets by two nozzles. To reduce the carryover of desiccant droplets, two demister were installed in the absorber and stripper, respectively. The solution is driven by a solution motor to flow to the inlet of the solution, where the solution is sprayed as atomized droplets by nozzles, and the droplets are dispersed homogeneously to contact air in the absorber. Air

is driven by an air compressor to flow through impinge for adjusting air humidity, and the moisture in the humid air is absorbed by the contact between liquid droplets and humid air. After the dehumidification, the dilute solution is pumped by a solution motor to the heat exchanger and to the heater to increase solution temperature. Then the stripping process is conducted to regenerate the solution in the stripper. The regenerated solution flows back to the absorber for reuse after the stripping process.

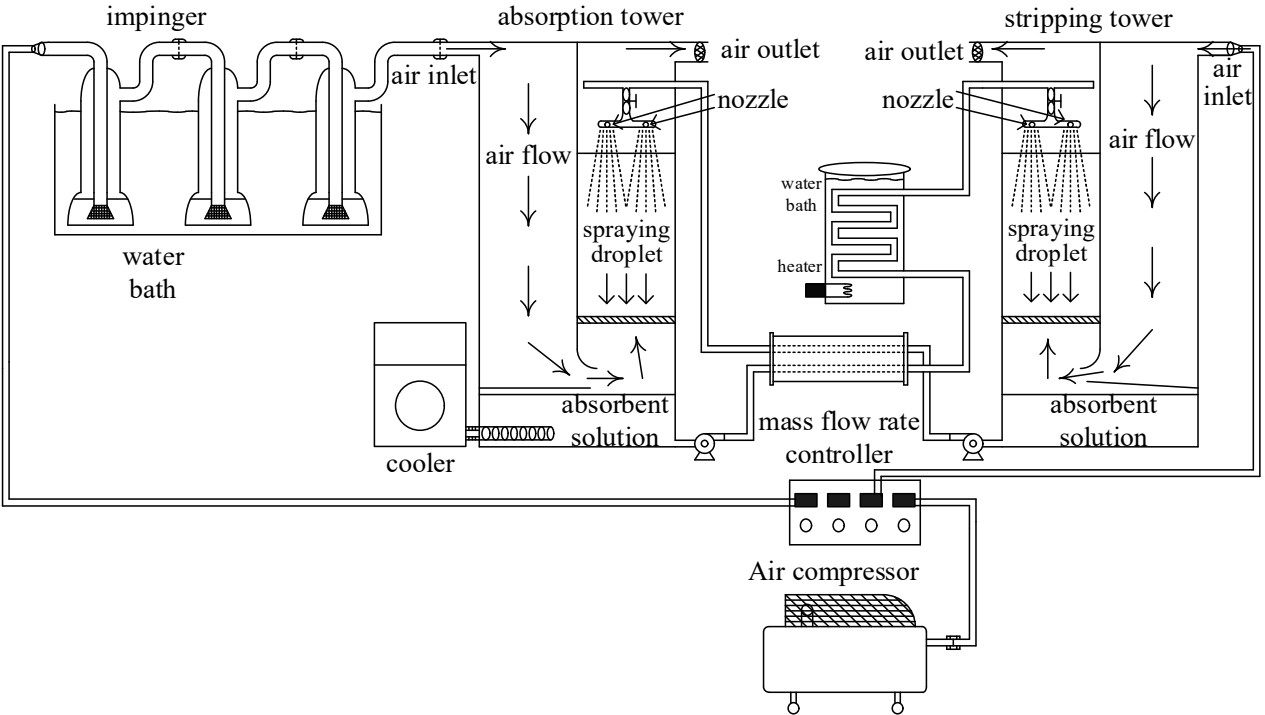

**Figure 2.** Spray-bed absorber in this study.

The desiccant solution used in this study is the TEG aqueous solution. The concentration of the TEG solution is in the range from 85 to 97.5 wt %, and the temperature is in the range from 20 to 35 °C. The concentration of the TEG solution was measured by a refractometer. The air flow rate treated by the spray-bed absorber is 495 to 842 L/min, and the desiccant flow rate is 5.16 to 15.81 L/min. The mass transfer coefficients and removal amount of water vapor are calculated based on the inlet, outlet, and operating conditions to discuss the effects of operating variables on mass transfer performance and to compare the data in the literature. The heat source for regenerating the solution was a 80-L insulated water tank with a 2-kW electric heater. Two TES-1160 hygrometers were used to measure the absolute humidity in the inlet and outlet of the absorber, respectively. The air flow rates were controlled by a mass flow controller, Model 5850E, and the TEG flow rates were controlled by a rotameter.

## 3. Results and Discussion

### 3.1. Vapor Pressure of the Desiccant Solution

When a desiccant solution is used to remove moisture from air, the moisture in the air is reduced due to the contact between the desiccant solution and water vapor, which causes a vapor pressure difference between the bulk air and the surface of the desiccant solution. The water vapor is guided to and absorbed by the desiccant solution due to the difference in vapor pressure. As mentioned above, the vapor pressures of the TEG solution were measured by the vapor pressure measuring system, and the results were shown in Figure 3. The difference between the vapor pressure measured in this study and that obtained from the DOW Chemistry Handbook [26] is less than 4%. The results demonstrated that the

measuring system of vapor pressure was set successfully in this study. Actually, the vapor pressure of the pure TEG solution approaches zero. The vapor pressure of the TEG solution contributed by water can be deduced. The more water in the solution, the higher the vapor pressure. The vapor pressure of the TEG solution increases with increases in the water component, as shown in Figure 3. In addition, the vapor pressures of the TEG solutions are increased with increases in the temperature.

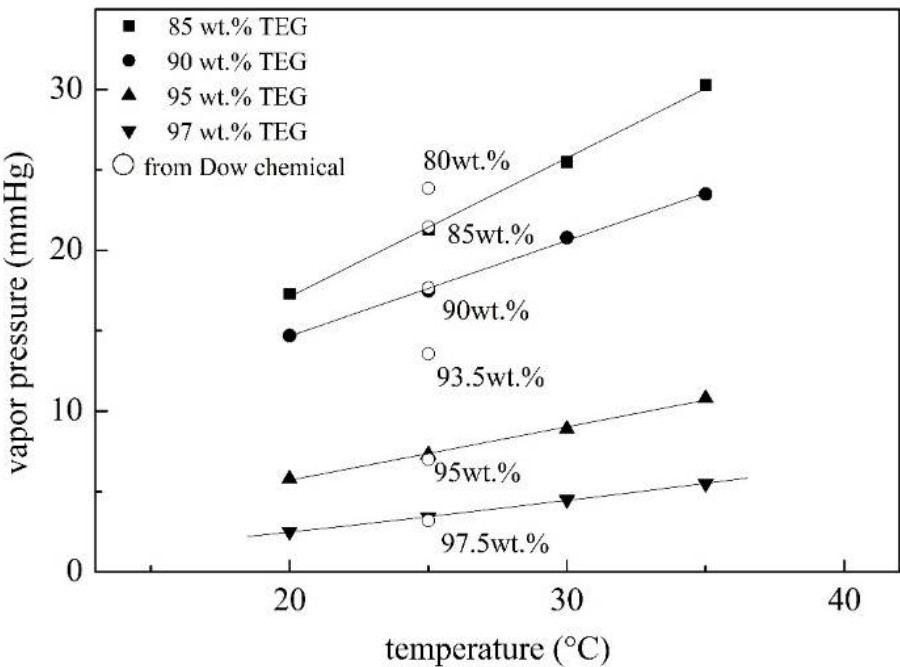

**Figure 3.** Vapor pressure of TEG aqueous solution.

### 3.2. Mass Transfer Performance Compared with Literature Data

To verify the reliability of the spray-bed absorber established in this study, the removal amounts of water vapor in this study were compared with literature data for the similar-scale packed-bed [27] and spray-bed [13] absorbers. As shown in Figure 4, all the studies show that the removal amount increases with increases in the liquid flow rate. Figure 4 also shows that the removal amount in this study and in the literature were not far apart, which establishes the success of this spray-bed absorber. In addition, the results confirm that the mass transfer performance of the spray-bed absorber is equivalent to that of the packed-bed absorber under maintaining a good spray effect.

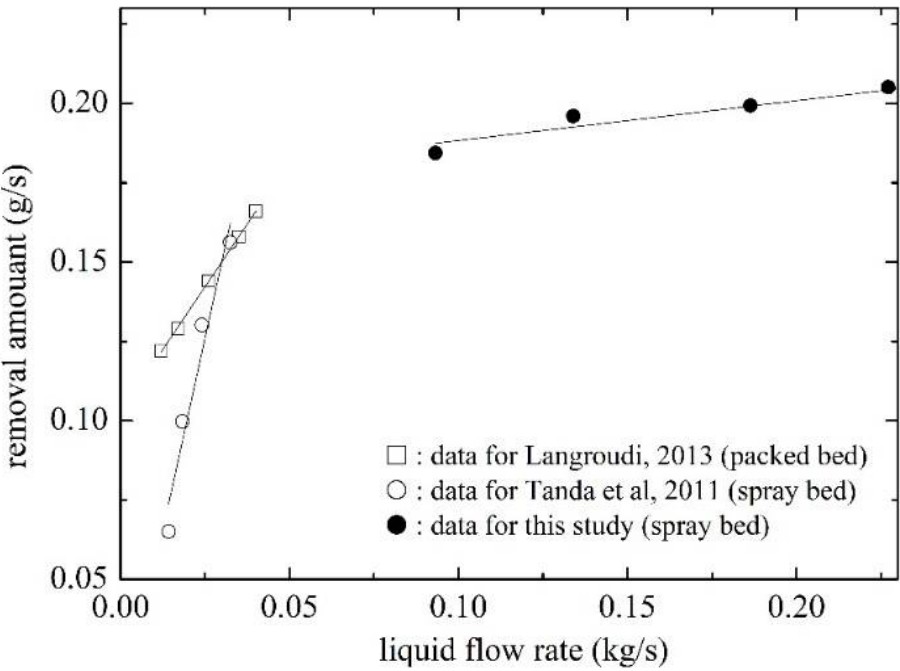

**Figure 4.** Removal amount compared with literature data.

### 3.3. Effects of Operating Variables on Mass Transfer Coefficient

The calculation of the volumetric mass transfer coefficient for the gas phase in the absorber can be referred from Hines et al. [28] or Geankoplis [29], and the formula for the mass transfer coefficient is shown as

$$K_{mA} = \frac{G}{Z} \int_{y_{Ain}}^{y_{Aout}} \frac{(1 - y_A^*)_M}{1 - y_A} \frac{dy_A}{y_A^* - y_A} \tag{1}$$

where

$$(1 - y_A^*)_M = \frac{(1 - y_A^*) - (1 - y_A)}{\ln\left[(1 - y_A^*)/(1 - y_A)\right]} \tag{2}$$

If the concentration of solute A in the gas phase is dilute, such as moisture in the air, the values of $(1 - y_A^*)_M$ and $(1 - y_A)$ will approach 1. The volumetric mass transfer coefficient for the gas phase can be simplified as

$$K_{mA} = \frac{V}{Z} \int_{y_{Ain}}^{y_{Aout}} \frac{dy_A}{y_A^* - y_A} \tag{3}$$

Therefore, the mass transfer coefficients for absorbing water vapor in the air were calculated based on Equation (3).

Figure 5 shows the effects of gas and liquid flow rates on mass transfer coefficient. The liquid flow rate was controlled at 12.59 L/min to discuss the coefficient changed with the gas flow rate, and the gas flow rate was controlled at 595 L/min to discuss the coefficient changed with the liquid flow rate. The amount of water vapor flowing into the absorber per unit time would increase with the larger gas flow rate, resulting in the more water vapor absorbed by the TEG solution. Therefore, Figure 5 shows that the mass transfer coefficients increase with increases in the gas flow rate. The desiccant solution can be regarded as the material for treating a substance, and the water vapor can be regarded as the substance to be treated. If the amount of the material for treating a substance is increased, the performance for treating the substance will naturally increase. Therefore, the mass transfer coefficient is increased with increases in the liquid flow rate, as shown in Figure 5.

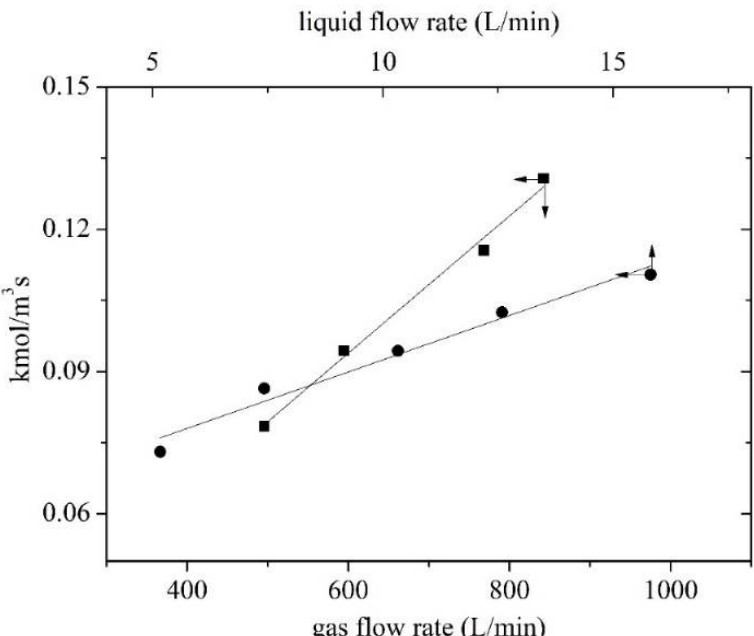

**Figure 5.** Effects of gas and liquid flow rates on mass transfer coefficient. (●: liquid flow rate; ■: gas flow rate).

Figure 6 shows the effect of concentration on mass transfer coefficient under the conditions of 595 L/min gas flow rate and 10.33 L/min liquid flow rate. The higher the concentration, the lower the vapor pressure, meaning that the larger the depression in vapor pressure. The higher driving force in the dehumidification process would be resulted from the larger depression in vapor pressure. Therefore, Figure 6 shows that the mass transfer coefficient for absorbing water vapor by the TEG solution is increased with increases in the TEG concentration, and the increases are more significant for the higher concentrations. Similarly, the lower the temperature of the TEG solution, the lower the vapor pressure. The higher driving force would be resulted from the lower temperature of the TEG solution in the dehumidification process. The mass transfer coefficient is decreased with increases in the temperature of the TEG solution, as shown in Figure 6.

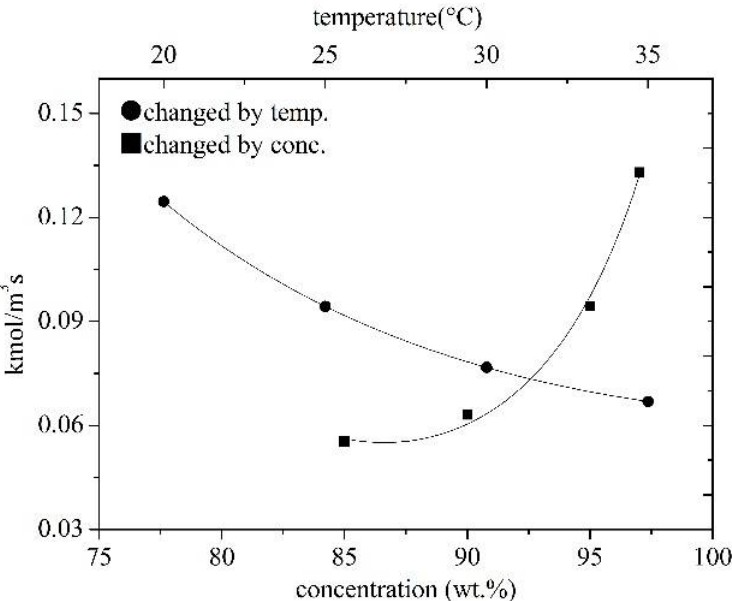

**Figure 6.** Effects of concentration and temperature of the TEG solution on mass transfer coefficient.

As described in the section of Introduction, the vapor pressure of the desiccant aqueous solution can be regarded as the depression in vapor pressure, and the larger depression is caused by the lower vapor pressure of the TEG solution. Therefore, the mass transfer performance would be associated with the inverse of the vapor pressure. Since the temperature and humidity of inlet air are almost unchanged to discuss mass transfer performance affected by operating variables, the partial pressures of water vapor are also changed insignificantly. To follow the dimensionless analysis, the ratio of the partial pressure of inlet water vapor to the vapor pressure of the TEG solution was considered. Figure 7 shows the ratio affected by the concentration and temperature of the TEG solution. Since the higher ratio is for the lower vapor pressure of the TEG solution, the higher ratio is for the higher concentration and the lower temperature of the TEG solution. Therefore, Figure 7 shows that the ratio is increased with increases in the concentration of the TEG solution and decreased with increases in the temperature of the TEG solution. While comparing Figure 7 with Figure 6, both the mass transfer coefficient and the ratio of pressure increase with increases in the concentration and decreases with increases in the temperature. In addition, the change of both is similar. The results can be attributed to the fact that the mass transfer coefficient would be associated with the ratio of pressure or the depression in vapor pressure. Figure 8 shows the relationship between the mass transfer coefficient and the ratio of the partial pressure of inlet water vapor to the vapor pressure of the TEG solution. Since the change in vapor pressure depression is caused by the concentration and temperature of the TEG solution, the depressions in discussing Figure 5 are almost unchanged due to the fixed concentration and temperature. Therefore, data in Figure 8 are focused on controlling the different concentration and temperature of the TEG solution. The result shows that the mass transfer coefficient is associated with the ratio of pressure and increased with increases in the ratio.

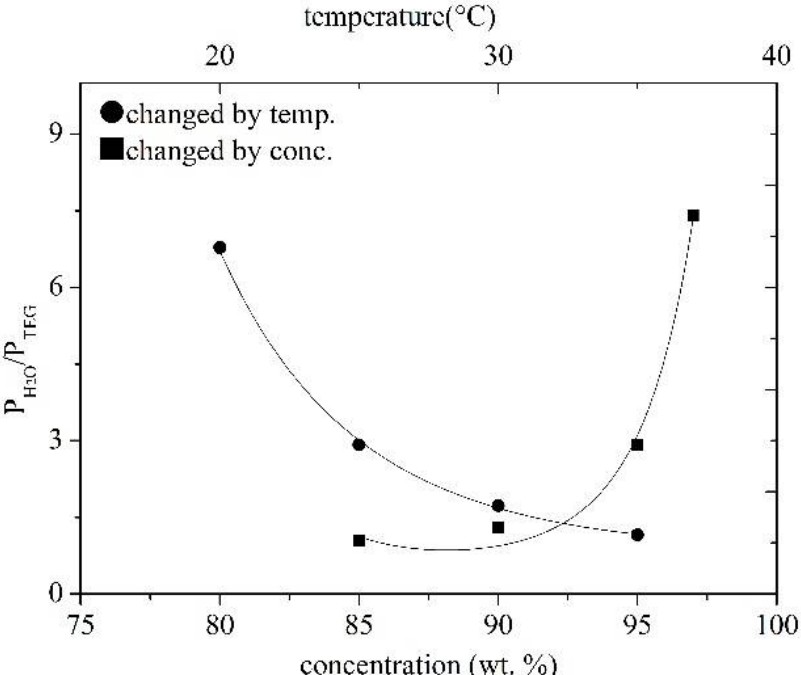

**Figure 7.** Effects of concentration and temperature of the TEG solution on the ratio of the partial pressure of water vapor to the vapor pressure of the TEG solution.

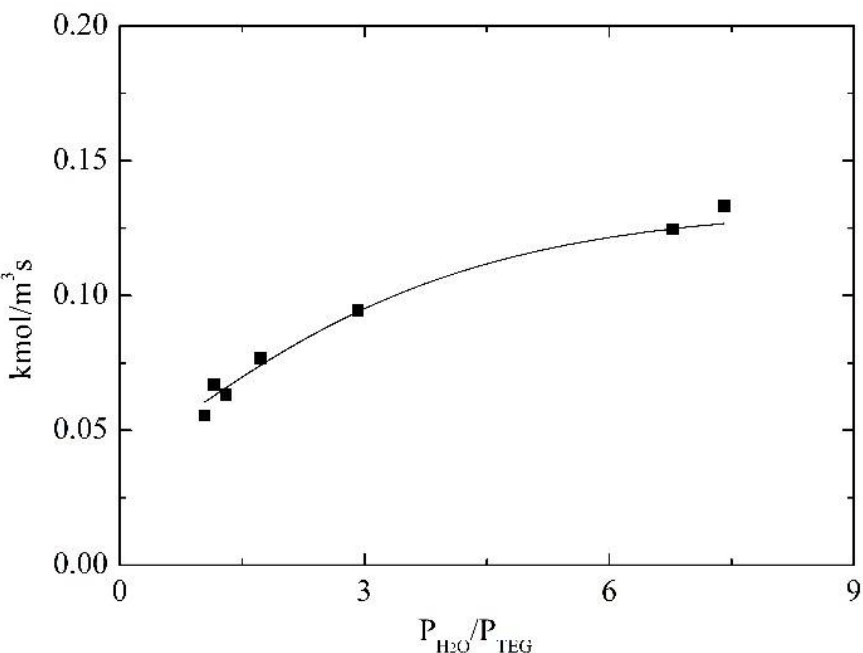

**Figure 8.** Effect of ratio of pressure on mass transfer coefficient. (■: data concerning the changed concentration and temperature of the TEG solution).

*3.4. Mass Transfer Correlation*

The mass transfer correlations were obtained by dimensional analysis and nonlinear regression in this study. Applying the dimensional analysis to reintegrate the fluid physical properties, the characteristics of the spray-bed absorber, and the relevant variables as a measurable dimensionless group. The fluid physical properties include the viscosity, the density, the diffusion coefficient, and the molecular weight. The column diameter and the gas–liquid interfacial area are the characteristics of the absorber, but the gas–liquid interfacial area is usually integrated into the volumetric mass transfer coefficient. The relevant variables include the gas and liquid flow rates and the depression in vapor pressure. The purpose to develop the mass transfer correlation is to predict the mass transfer performance of the system before conducting experimental runs. Therefore, all the variables and the characteristics mentioned above are known, and integrated to form a dimensionless correlation.

$$\frac{K_{mA}Md_c^2}{\rho D} = \alpha\left(\frac{\rho v d_c}{\mu}\right)^{\beta}\left(\frac{\mu}{D\rho}\right)^{\gamma}\left(\frac{L}{G}\right)^{\delta}\left(\frac{P_{H2O}}{P_{TEG}}\right)^{\eta} \tag{4}$$

The mass transfer coefficients were obtained from experimental runs and all the variables and characteristics were brought into Equation (4), and the nonlinear regression was conducted to obtain the values for $\alpha$, $\beta$, $\gamma$, $\delta$, and $\eta$. Finally, the mass transfer correlations were obtained as

$$\frac{K_{mA}d_c^2}{\rho D} = 0.0147Re^{1.45}Sc^{0.35}\left(\frac{L}{G}\right)^{0.24}\left(\frac{P_{H_2O}}{P_{TEG}}\right)^{0.44} \tag{5}$$

The relationship between the experimental mass transfer coefficient and the predicted values from Equation (5) are represented as black dots in Figure 9. The R-square is 0.997 and the average error is within 5%. The vapor pressure of the TEG solution depends on the concentration and the temperature. If the term, the ratio of pressures, is replaced by the concentration and the temperature, a dimensionless correlation will be shown in Equation (6).

$$\frac{K_{mA}Md_c^2}{\rho D} = \alpha Re^{\beta}Sc^{\gamma}\left(\frac{L}{G}\right)^{\delta}\left(\frac{T_s}{273}\right)^{\eta}X^{\lambda} \tag{6}$$

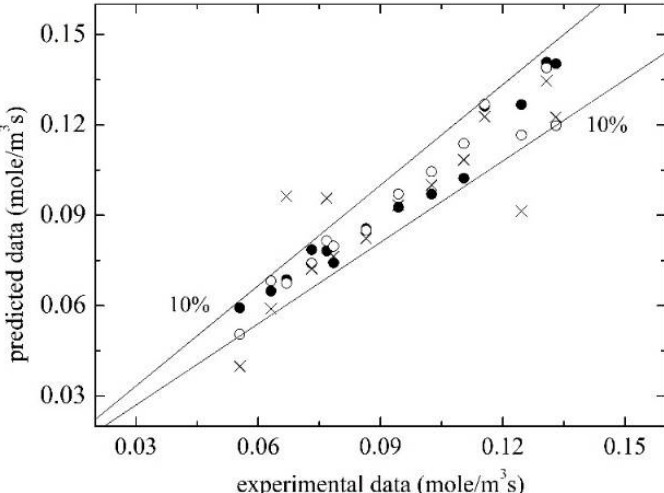

**Figure 9.** Relationship between experimental and predicted mass transfer coefficient. (●: predicted by Equation (5); ○: predicted by Equation (7); ✕: predicted by Equation (9)).

The unknown parameters were obtained by nonlinear regression as

$$\frac{K_{mA}Md_c^2}{\rho D} = 0.178Re^{1.45}Sc^{0.35}\left(\frac{L}{G}\right)^{0.39}\left(\frac{T_s}{273}\right)^{-10.2}X^{1.22} \tag{7}$$

The relationship between the experimental mass transfer coefficient and the predicted values from Equation (7) are represented as the circle in Figure 9. The R-square is 0.997 and the average error is also within 5%. Although the accuracies for Equation (5) and Equation (7) are similar in predicting the mass transfer coefficient, there are just two terms associated with the operating variables, that is $L/G$ and $P_{H_2O}/P_{TEG}$, in Equation (5) to predict the mass transfer coefficient, except for the fluid physical properties and the characteristics of the absorber, all the operating variables, such as the gas and liquid flow rates and the concentration and temperature of the TEG solution, were used in Equation (7) to predict the mass transfer coefficient.

To simply the mass transfer correlation, the temperature and the concentration are integrated as Equation (8).

$$\frac{K_{mA}Md_c^2}{\rho D} = \alpha Re^{\beta}Sc^{\gamma}\left(\frac{L}{G}\right)^{\delta}\left(\frac{T_s/273}{X}\right)^{\eta} \tag{8}$$

The unknown parameters were obtained by nonlinear regression as

$$\frac{K_{mA}Md_c^2}{\rho D} = 0.084Re^{1.45}Sc^{0.35}\left(\frac{L}{G}\right)^{0.37}\left(\frac{T_s/273}{X}\right)^{1.58} \tag{9}$$

The relationship between the experimental mass transfer coefficient and the predicted values from Equation (9) are represented as the symbol ✕ in Figure 9. The R-square is 0.982 and the average error is about 10%. The more data points beyond the deviation 10% in using Equation (9) to predict the mass transfer coefficient. The effects of concentration and temperature on mass transfer coefficient are different. However, the only one exponent in the integrated term, $(T/273)/X$, means that the effects of concentration and temperature are the same, which leads to the larger error in predicting the mass transfer coefficient.

Finally, the mass transfer data in the literature [13] were used to test the mass transfer correlation developed in this study. Since the operating conditions are different from this study, the degree for operating variables affecting the mass transfer coefficient is different naturally. For example, the increase of mass transfer coefficient with increases in the concentration of the TEG solution was proven in this study; however, the decrease of

mass transfer coefficient with increases in the concentration of the PEG solution was found by Tanda et al. [13]. Since the density and viscosity of the PEG solution increase with increases in the concentration of the PEG solution, the narrower spray angle accompanies with the higher concentration of the PEG solution, which results in the lower mass transfer coefficient. The spray angle is almost the same in this study due to the larger liquid flow rate. There are two advantages to develop the mass transfer correlation containing the ratio of the partial pressure of inlet water vapor to the vapor pressure of the TEG solution. One is that the vapor pressure depends on the concentration and the temperature. The other is that the depression in vapor pressure implies the concept of the driving force in the dehumidification process. The result from literature data [13] seems to reveal that the effect of spray angle is stronger than that of driving force on the mass transfer coefficient under the lower liquid flow rate. In general, the higher the concentration of the desiccant solution, the greater the driving force for the dehumidifier. Although the proposed result is inconsistent with the common concept, the ratio of pressures can still be hold in the mass transfer correlation to show the effect of concentration on the performance. The dimensionless method and nonlinear regression were conducted to obtain the mass transfer correlation as

$$\frac{K_{mA}Md_c^2}{\rho D} = 0.00111 Re^{1.45} Sc^{0.35} \left(\frac{L}{G}\right)^{0.95} \left(\frac{P_{H_2O}}{P_{TEG}}\right)^{-0.33} \tag{10}$$

The negative exponent on the ratio of pressures indicates that the mass transfer coefficient increases with decreases in the concentration of the desiccant solution. Figure 10 shows the relationship between the experimental data by [13] and the predicted values from Equation (10). Comparison of mass transfer data in this study with that from [13] also provides in Figure 10. The relationship between the experimental mass transfer coefficient and the predicted values from Equation (10) are represented as the circle in Figure 10. The R-square is 0.992 and the average error is about 7%. The result demonstrates that the mass transfer coefficient from literature data could also be predicted accurately by the mass transfer correlation including the depression in vapor pressure. The volumetric mass transfer coefficient depends on the gas flow rate, as shown in Equation (1). The mass transfer coefficient is greater for this study could be attributed to the fact that the larger gas flow rate is operated in this study.

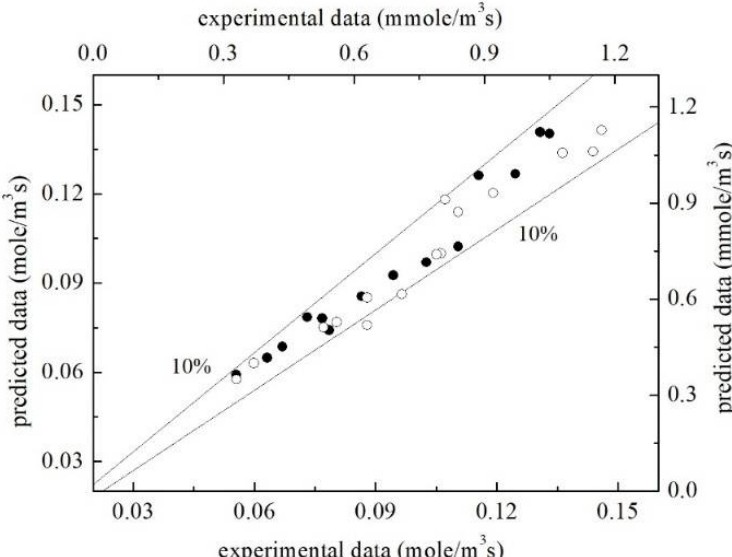

**Figure 10.** Regression of literature data and Comparison with this study. (●: dada in this study corresponding to the left and upper axes; ○: data from literature [13] corresponding to the right and bottom axes).

## 4. Conclusions

The depression in vapor pressure caused by adding desiccant solution into liquid water is the driving force in the dehumidification process. The vapor pressure depends on the temperature and the concentration. The mass transfer coefficient affected by operating variables and the mass transfer correlation developed by the depression in vapor pressure were discussed in this study. The experimental results show that the mass transfer coefficients increase with increases in the gas and liquid flow rates. The increases of the mass transfer coefficient and the ratio of pressures with increases in the concentration and the decreases of those with increases in the temperature were shown in this study. The relationship between the ratio of the partial pressure of inlet water vapor to the depression in vapor pressure and the mass transfer coefficient proves that the mass transfer coefficient is associated with the ratio of pressures and increased with increases in the ratio. Finally, the characteristics of the spray-bed absorber, and the relevant variables, were reintegrated by the dimensional analysis and the mass transfer correlations were obtained by the non-linear regression. Except for the fluid physical properties and the characteristics of the absorber, the accuracies between the mass transfer correlation involving $P_{H_2O}/P_{TEG}$ and that involving $T/273$ and $X$ are similar in predicting the mass transfer coefficient. The result can be attributed to the fact that the mass transfer coefficient would be associated with the ratio of pressures or the depression in vapor pressure. Since the term $P_{H2O}/P_{TEG}$ depends on the concentration and the temperature, the effect of concentration and temperature on the mass transfer coefficient can be represented by vapor pressure. The correlation tested by literature data shows that the ratio of pressures can still be used in the mass transfer correlation under the condition of the mass transfer performance decreased with increases in the concentration.

**Author Contributions:** Conceptualization: H.-T.W. and C.-C.C.; Experimental runs and analysis: H.-T.W. and C.-C.C.; Writing: H.-T.W. and C.-C.C. All authors have read and agreed to the published version of the manuscript.

**Funding:** This research received no external funding.

**Institutional Review Board Statement:** Not applicable.

**Informed Consent Statement:** Not applicable.

**Data Availability Statement:** The vapor pressures of the TEG solution were referred from Dow Chemical Company, Triethylene Glycol, https://pdf4pro.com/view/triethylene-glycol-dow-57b1b2 .html (accessed on 21 June 2021).

**Conflicts of Interest:** The authors declare no conflict of interest.

## Nomenclature

**Symbols**
$A_t$: surface area of the packing per unit volume of the bed, $1/m$
$c$: molar concentration ($mol/m^3$)
$C_v$: mass-transfer factor, Dimensionless
$D$: diffusion coefficient, $m^2/s$
$d_c$ = column diameter, m
$d_e$: equivalent diameter, $4\varepsilon/a_d$, m
$d_n$: nozzle diameter, m
$G$: gas flow rate, L/min
$Gr$: Grashof number, dimensionless
$h_k$: local mass transfer coefficient, $kg/m^2s$
$H_{kA}$: overall mass transfer coefficient, $kg/m^3s$
$h_L$: Liquid holdup, Dimensionless
$h_m$: mass transfer coefficient, $g/m^2s$

$H_{mA}$: overall mass transfer coefficient, $g/m^3s$
$k_k$: local mass transfer coefficient, $kgmole/m^2s$
$k_m$: local mass transfer coefficient, $mole/m^2s$
$K_{mA}$: overall mass transfer coefficient, $mole/m^3s$
$K_W$: wall factor, dimensionless
L: liquid flow rate, L/min
$L_s$: superficial liquid mass velocity, $kg/m^2s$
M: molecular weight of water, kg/mol
m: mass flow rate, kg/s
P: pressure, atm
$P_{H2O}$: partial pressure of water pressure, mmHg
$P_{TEG}$: vapor pressure of TEG solution, mmHg
R: ideal gas constant, $m^3atm/kmol \cdot K$
Re: Reynolds number, $\rho vdc/\mu$
S: superficial mass velocity, $kg/m^2s$
Sc: Schmidt number, $\mu/D\rho$
Sh: Sherwood number, dimensionless
T: Temperature, °C
$T_s$: solution temperature, °C
v: fluid flow velocity, m/s
V: gas molar flow rate, $kmole/m^2s$
X: concentration in mole fraction
$y_{Ain}$: molar ratio of component A in inlet gas
$y_{Aout}$: molar ratio of component A in outlet gas
$y_A^*$: molar ratio of component A at equilibrium state
Z: height of the absorber
**Greek Symbols**
$\varepsilon$: packing porosity or void fraction, dimensionless
$\alpha$: regressed parameter for the correlation
$\beta$: regressed parameter for the correlation
$\gamma$: regressed parameter for the correlation
$\delta$: regressed parameter for the correlation
$\eta$: regressed parameter for the correlation
$\mu$: fluid viscosity, kg/ms
$\rho$: fluid density, $kg/m^3$
$\sigma$: surface tension, $kg \cdot m/s^2$
**Subscript**
a: air
G: gas phase
L: liquid phase
s: solution
w: water

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
