# Peer review of "Effects of Vapor Pressure of Desiccant Solution on Mass Transfer Performance for a Spray-Bed Absorber"

_processes, doi:10.3390/pr9091517_

Round 1
Reviewer 1 Report
Dear Authors
The article is well arranged, the layout is classic. The results presented are fully discussed and conclusions are drawn from the research results. However, the article has a few inaccuracies:
- The description of the measurement steps is superficial and must be extended with further explanations:
- how long and how desiccant solution was cooling;
- where liquid nitrogen is placed and what temperature should desiccant solution have?
- explain step 3: in what moment valves V1, V3, V5 and V6 are opening, what is happening in this moment with desiccant solution? What is the purpose of opening these valves?
- in my opinion steps 4 and 5 have conflicting information. In what moment valve V1 is opening? How the system reaches the vacuum state as valve V1 is closed?
- what is happening with desiccant solution after cooling and opening the valves V1, V3, V5 and V6;
- at what point is the vapour created?
- Figure 2 should be complete with a description of the components on the figure. Where are Inlet/outlet, tower, nozzles, where is spraying takes place and other. Authors should show direction of solution in progress.
- Figure 9 it must be moved up, right after the paragraph describing and discussing this Figure.
Reviewer 2 Report
Article: processes-1337656
Title: Effects of Vapor Pressure of Desiccant Solution on Mass Transfer Performance for a Spray-Bed Absorber
Authors: Hung-Ta Wu and Chin-Chun Chung
The authors propose mass transfer correlations to predict the effect of the vapor pressure on dehumidification process performance using water solutions containing TEG as desiccants. The authors provide an extensive review of the state of art. They also deliver a systematic dimensionless analysis and contribute new data from trimethylene glycol (TEG) solutions. The article is complete and well written.
I recommend publishing the article after a few minor corrections:
Improve the image quality of the figures.
Table 1: the acronym COP is not clarified throughout the text; consider adding its meaning in the legend of the table.
Lines 239-240: ‘The difference between the vapor pressure measured in this study and obtained in the Chemistry Handbook [26] is less than 4%.’. Consider rephrasing The difference between the vapor pressure measured in this study and that obtained from the DOW Chemistry Handbook [26] is less than 4%.’
Line 265: change the name Geankoplies; it is Geankoplis.
